# Linking Benzene, in Utero Carcinogenicity and Fetal Hematopoietic Stem Cell Niches: A Mechanistic Review

**DOI:** 10.3390/ijms24076335

**Published:** 2023-03-28

**Authors:** Nur Afizah Yusoff, Zariyantey Abd Hamid, Siti Balkis Budin, Izatus Shima Taib

**Affiliations:** Center for Diagnostic, Therapeutic and Investigative Studies (CODTIS), Faculty of Health Sciences, Universiti Kebangsaan Malaysia, Kuala Lumpur 50300, Malaysia

**Keywords:** benzene, fetal, in utero, hematopoietic stem cells and progenitors, carcinogenicity, oxidative stress, epigenetic, chromosome aberration, hematological disorders

## Abstract

Previous research reported that prolonged benzene exposure during in utero fetal development causes greater fetal abnormalities than in adult-stage exposure. This phenomenon increases the risk for disease development at the fetal stage, particularly carcinogenesis, which is mainly associated with hematological malignancies. Benzene has been reported to potentially act via multiple modes of action that target the hematopoietic stem cell (HSCs) niche, a complex microenvironment in which HSCs and multilineage hematopoietic stem and progenitor cells (HSPCs) reside. Oxidative stress, chromosomal aberration and epigenetic modification are among the known mechanisms mediating benzene-induced genetic and epigenetic modification in fetal stem cells leading to in utero carcinogenesis. Hence, it is crucial to monitor exposure to carcinogenic benzene via environmental, occupational or lifestyle factors among pregnant women. Benzene is a well-known cause of adult leukemia. However, proof of benzene involvement with childhood leukemia remains scarce despite previously reported research linking incidences of hematological disorders and maternal benzene exposure. Furthermore, accumulating evidence has shown that maternal benzene exposure is able to alter the developmental and functional properties of HSPCs, leading to hematological disorders in fetus and children. Since HSPCs are parental blood cells that regulate hematopoiesis during the fetal and adult stages, benzene exposure that targets HSPCs may induce damage to the population and trigger the development of hematological diseases. Therefore, the mechanism of in utero carcinogenicity by benzene in targeting fetal HSPCs is the primary focus of this review.

## 1. Introduction

Benzene toxicity has emerged as a significant health issue. Benzene, C_6_H_6_, is a colorless and flammable sweet-smelling liquid. It easily evaporates into the air and has a low solubility in water. Natural and industrial sources both contribute to the production of benzene, a chemical with numerous applications. Benzene is a group 1 carcinogen, which can cause cancer in both humans and animals [1]. It is present in the environment and is used in many industries. Benzene exposure to the public occurs through cigarette smoking, gasoline vapors and car exhaust, as well as water and soil that are contaminated with benzene [2]. Meanwhile, industries throughout the world rely on it as a versatile solvent or precursor in a wide range of products, including plastics, polymers, detergents, pesticides, rubbers, dyes, medicines and explosives [3,4,5]. Hence, the human population is vulnerable to environmental and occupational benzene exposure.

Exposure to benzene chronically induces depression in bone marrow, which commonly presents as clinically reduced counts in peripheral blood cell (leukopenia, anemia, thrombocytopenia and/or pancytopenia) as well as other hematological malignancies such as myelodysplasia or leukemia [4,6]. According to Ross and Zhao [7], the toxicity of benzene inflicted on bone marrow is significantly influenced by benzene’s metabolism. Bone marrow is the main site for HSCs niches that maintain hematopoiesis [8]. Meanwhile, both cell intrinsic (e.g., transcriptional factors) and cell extrinsic (e.g., cytokines) factors control the self-renewal and differentiation of HSCs and lineage-committed progenitors into mature and functional blood cells inside the niche [9]. Therefore, it is essential that HSCs and progenitor cells maintain a delicate balance between self-renewal and differentiation in order to prevent alterations in hematopoiesis and potentially, the development of hematological malignancies [10]. Although there are worrying reports of the genotoxic effects of benzene targeting the bone marrow microenvironment and hematopoietic system, closer scrutiny shows that these reports were primarily focused on benzene exposure and its effects that target only a selective subpopulation of cells. Blood lymphocytes, Sca-1^+^ and CD34^+^ cells are among the most common subpopulation of cells being studied, which do not represent the complex microenvironment of the HSC niche which is comprised of lineage-specific subpopulation cells that are vital for the maintenance of hematopoiesis [11].

The lineage-directed strategy has been found to have an important part in mediating the benzene toxicity effects on adult stem cells, specifically in the HSC and multilineage hematopoietic stem and progenitor cell (HSPCs) niche. In exposed HSPCs, benzene causes genotoxicity, leukemogenicity and hematotoxicity by inducing chromosome aberrations [12], oxidative stress and apoptosis [4,12], aberrant deoxyribonucleic acid (DNA) repair mechanisms and epigenetic alterations [13], DNA damage [4] and changes in gene expression regulating self-renewal and differentiation [12]. Recently, our group has established a lineage-dependent response based on emerging data linking HSPCs of benzene toxicity on different lineages [14]. These findings show that upon 1,4-benzoquinone (1,4-BQ) exposure, the clonogenicity of myeloid progenitors is selectively inhibited as compared with lymphoid progenitors, suggesting the role of lineage-specificity in governing benzene toxicity targeting HSPC niche. Additionally, a previous study discovered that non-cytotoxic concentrations of 1,4-BQ exposure modifies the fate of HSPCs by altering the genes regulating the self-renewal and differentiation pathway [12]. Moreover, to the best of our knowledge, our study was the first to report a mechanism involving lineage-specific hematopoietic cells subpopulation in mediating benzene toxicity targeting the HSPCs niche for erythroid, Pre-B lymphoid and myeloid progenitors [14]. Thus, it is essential to explore similar mechanisms of benzene toxicity toward the HSPC niche in the fetal stage.

The intricacy of the HSPC milieu is a big challenge in understanding the specific mechanism linking benzene to the dysregulation of the HSPC niche. Nevertheless, there have been multiple reports on the effects of benzene toxicity on bone marrow and HSPCs. Previous studies have reported some evidence of benzene toxicity affecting HSPCs’ self-renewal and differentiation pathways that led to the clonal expansion of leukemic stem cells (LSCs) [15]. It is well known that for the malignant properties of leukemias, the acquisition of self-renewal by LSCs is required. Despite the numerous studies investigating the mechanism of leukemogenesis, there remains a knowledge gap for further exploration particularly concerning childhood leukemogenesis.

Generally, carcinogens exhibit ≥1 of the 10 principal characteristics that induce cancer [16] as shown in Table 1. These effects can happen at low benzene exposure levels in humans (≤1 parts per million (ppm)). Workers exposed to benzene have an increased risk of developing leukemia [17]. So far, no case of direct occupational exposure to benzene among children have yet been reported. Nevertheless, a few recent epidemiological findings have suggested that benzene exposure in the environment may be a significant contributor to the prevalence of childhood leukemia. Moreover, epidemiological studies have documented the toxic consequences of maternal benzene exposure during pregnancy on the unborn fetus. For instance, according to Freedman et al. [18], pregnant mothers that are exposed to interior house painting in the year leading up to their children birth are at an increased risk of toxicity exposure. The placenta controls the flow of nutrients from the mother to the developing embryo throughout pregnancy [19]. As a result, maternal exposure to leukemogenic substances during early pregnancy may cause enhanced DNA instability and genetic vulnerability to cancer development of fetus, resulting in the development of childhood leukemia in the offspring [20]. To date, studies concerning benzene toxicity targeting bone marrow are mostly associated with adult systems and only limited studies on fetal systems have been described, indicating the need for further exploration. A number of potential underlying mechanisms that mediate fetal bone marrow toxicity by benzene have been reported. During the fetal stage, HSPCs are in the active division, in which the central function of HSPCs is to rapidly maintain the homeostatic blood cells level in the growing fetus for oxygen transport and immune system development, thus making them a preferential target of benzene [21]. Furthermore, in adult hematopoiesis studies, reports suggest that the presence of an intrinsic self-protecting machinery in HSPCs in adults can protect the stem cell compartment in the hematopoietic system [22]. Hence, it is tempting to speculate that the absence of self-protecting mechanisms in fetal HSPCs renders the cells susceptible to malignant transformation [23]. The high level of the myeloperoxidase (MPO) enzyme in bone marrow plays a crucial role in the hydroquinone (HQ) oxidization to a more stable but toxic metabolite known as p-benzoquinone (p-BQ) or 1,4-BQ, which is believed to be involved in benzene-mediated carcinogenicity [24]. However, the role of MPO in governing benzene toxicity targeting fetal bone marrow remains unexplored and deserves further investigation.

Therefore, due to accumulating evidence linking benzene to childhood blood malignancies such as leukemia, the main objective of this review is to discuss the mechanism of benzene-induced in utero carcinogenicity, focusing on the toxicity outcomes targeting the fetal HSPC niche consisting of lineage-specific hematopoietic progenitors, which are the common myeloid progenitors (CMP) and common lymphoid progenitors (CLP).

This review is structured as follows. “Survey Methodology” elaborates on the search criteria we used to find articles and references. “Hematopoiesis” comprises “Hematopoiesis in the Adult Phase” which elaborates the hematopoiesis maintenance regulated by the roles of HSCs and lineage-specific hematopoietic progenitors, and “Hematopoiesis in the Fetal Phase”, which discusses the different waves of hematopoiesis that occurs in different fetal sites. “Benzene Metabolism Linked to Fetal Toxicity via Maternal and Paternal Exposure” discusses the benzene metabolism and biological mechanisms of childhood cancer in relation to exposure time windows via maternal and paternal exposures. “Mechanisms of Benzene-induced in Utero Carcinogenicity involving Hematopoietic Stem Cells and Multilineage Progenitors” provides mechanistic information on how in utero exposure of benzene can affect HSPCs. “Covalent Binding” discusses the relationship of covalent binding and how it exerts toxicity on macromolecules. The effects of imbalance between reactive oxygen species (ROS) and oxidative damage are provided in “Oxidative Stress”. The genomic instability due to DNA damage following in utero exposure to benzene is discussed in “Error in DNA Repair Pathways”. “Chromosomal Aberration and Genetic Damage” reviews the ability of benzene to cause micronuclei and DNA damage. “DNA Methylation” as well as “Histone Modification and Chromatin Remodeling” which play important roles in fetal developments are discussed in the “Epigenetic Modification” subtopic. “Placenta-mediated Toxicity” discusses how trophoblast barriers can release factors that harm DNA in fetal stem cells. “The Origin of Hematological Diseases from in-utero Benzene Exposure” reviews the occurrence of fetal leukemogenesis due to maternal exposure to benzene. Finally, the conclusion and future remarks of this study are presented in “Conclusion and Future Remarks”.

## 2. Survey Methodology

Table 2 summarizes the article selection criteria used to construct this review. Briefly, journal databases, primarily Google Scholar and PubMed, were used to research the scholarly articles reviewed in this paper. The keywords used to search for these articles included “benzene”, “fetal”, “in utero”, “hematopoietic stem cell and progenitors”, “carcinogenicity”, “oxidative stress”, “epigenetic”, “chromosome aberration”, and “hematological disorders”. The inclusion criteria for the selected articles required the articles to be related directly to in utero exposure of benzene and its mechanism of toxicity, hematopoietic stem cells and progenitors and development of hematological disease. The searches were not refined by publishing date, authors, author affiliations, journals or the impact factors of the journals. The quantitative articles provided measurable data from experimental and epidemiological studies that revealed patterns and trends in carcinogenic effects due to benzene exposure. The qualitative articles provided insights into the problems and ideas or hypotheses underlying the carcinogenicity following maternal exposure to benzene. In summary, this review is based on 199 references that are comprised of 87 original research articles, 103 review articles, one conference paper, one commentary paper, six webpages and one thesis. These references focus on the underlying mechanisms of in utero exposure during maternal stage to benzene and the carcinogenicity effect primarily on fetal HSPCs.

## 3. Hematopoiesis

The mammalian hematopoietic system is remarkably efficient in meeting an organ-ism’s vital needs, yet is highly sensitive and finely regulated. Hematopoiesis is known to be regulated at different microenvironments depending on the developmental stage of either the adult or fetal phases. In brief, adult hematopoiesis mainly occurs at the bone marrow niche while fetal hematopoiesis begins in several waves throughout the body. Therefore, the following sections will provide an in-depth description of hematopoiesis that occurs in both adult and fetal phases. 

### 3.1. Hematopoiesis in the Adult Phase

Definitive erythropoiesis in postnatal life begins in the bone marrow and takes place under normal physiologic conditions. According to Nandakumar et al. [25], trabecular and spongy bone in infants produces erythrocytes while only the sternum, ribs, vertebra and proximal ends of long bones produce erythrocytes in adults. All adult hematopoietic cells are produced by HSCs in the bone marrow through a sequence of intermediate progenitors [25]. The central vein drains highly vascularized bone marrow, where arteries and arterioles enter the tissue and change into a sinusoidal network [26]. It creates the ideal microenvironment for stem cells to survive, regenerate themselves and then differentiate into specific lineage-committed progenitors and mature blood cells [21]. Large structures that define the microenvironment include a vascular network made up of arterioles that penetrate the bone and produce a sinusoidal network that drains through a central vein, and a reticular stromal cells network that covers around the various vessels and bone tissue that encloses the marrow. According to Wei and Frenette [27], these structures work together and regulate each other to keep the tissue functioning.

Daily, one trillion new cells emerge in the adult human bone marrow, making blood one of the most regenerating tissues [28]. The myeloid and lymphoid lineages are the two main branches that make up the hematopoietic system. Three crucial processes are carried out by the myeloid lineages such as oxygen transportation, hemostasis and innate immunity. Myeloid progenitors give rise to erythrocytes, which in turn facilitate efficient oxygen delivery throughout the body. Platelets, which coagulate blood, are produced by megakaryocytes in the myeloid lineage [29,30,31]. Innate immune cells, which are comprised of monocytes, neutrophils, eosinophils, basophils and dendritic cells, are made by the myeloid lineage and provide a broad, general defense against invading pathogens while also facilitating the establishment of adaptive immune responses. Meanwhile, adaptive immunity is produced by B, T and natural killer (NK) cells, which are produced by the lymphoid lineage [32,33].

Simply put, hematopoiesis has extensively been regarded as a standard of stem cell biology and tissue homeostasis in which it is estimated that an adult person creates 4–5 *×* 10^11^ hematopoietic cells every day [34]. Stem cell renewal and multipotent progenitor differentiation into mature blood cells are two stages of a complex process governed by a unique genetic blueprint that begins with asymmetrical cell division [10,35]. HSPCs are a population of immature cells that set themselves apart from other hematopoietic cells through their pluripotent capacity for self-renewal and differentiation. Two important HSPC niches that play prominent roles in the homeostatic regulation of HSPCs have been identified, which are vascular niche, as described earlier, and osteoblastic niche. Mesenchymal stem cells (MSCs) in bone marrow give rise to osteoblastic cells [36]. Bone marrow MSCs are adult, fibroblast-like multipotent cells that are essential in maintaining the microenvironment and tissue homeostasis. Human mesenchymal stem and progenitor cells (MSPCs) can be derived from the bone marrow, umbilical cord blood, placenta or adipose tissue [37]. In the bone marrow, MSPCs are involved in structure formation and organization of the hematopoietic microenvironment [38]. In patients with myelodysplastic syndromes (MDS), MSCs show a decreased expression of certain cell surface molecules [39], especially those involved in the interaction with HSPCs [40], including the adhesion molecules CD44 and CD49e (α5-integrin). MDS are malignant HSPC disorders that have the capacity to progress to acute myeloid leukemia (AML). The involvement of MSCs in benzene toxicity targeting bone marrow has been reported: a benzene metabolite, HQ reduces cell viability and induces apoptosis in bone marrow MSCs by downregulating the expression of multidrug resistance (MDR1) via a mechanism involving the nuclear factor kappa B (NF-κB) pathway, which could lead to hematotoxicity and leukemogenicity [41]. In conclusion, adult hematopoiesis is regulated by a complex microenvironment to ensure its homeostasis. In the following section, this review will further address hematopoiesis in the fetal stage.

### 3.2. Hematopoiesis in the Fetal Phase

Childhood cancer and the birth of children with genetic abnormalities are both thought to be brought on by disturbances in cellular differentiation or division. Hence, fetal well-being is an important aspect for health monitoring, including the preventive aspect of maternal and paternal exposures to genotoxic and carcinogenic agents. Stem cells contribute significantly to medical field not only for therapeutic purposes but also are important research subjects to elucidate their role in the pathogenesis of diseases, including those concerning fetal diseases or fetal origins of adult diseases. In the context of fetal hematopoiesis, a report using the mice model by Kikuchi and Kondo [42] states that adult stem cells will replace fetal stem cells following the first 2–3 weeks of life.

Contributing to the adult HSPCs, fetal hematopoiesis has been reported to be a time-dependent and a microenvironment-controlled process [43]. Hematopoiesis in the fetus is more complex because the anatomical site and state of energy production are highly dynamic and are dependent on the stage of gestational development and anatomical location [44]. During ontogeny, the developing HSPCs are required to complete a maturation process that permits their engraftment and maintenance in the future hematopoietic niche [44]. In addition, the initial miniature HSPCs pool that emerges from hemogenic sites must expand to establish an adequate supply of HSPCs for postnatal life. Eventually, the largely cycling fetal HSPCs progress into a quiescent state and maintain homeostasis in the adult [45,46]. During embryogenesis, the migration of fetal hemopoiesis through multiple sites is hypothesized to allow different inductive signals from the microenvironments to support the development of HSPCs. Fetal hematopoietic progenitor cells (HPCs) and HSCs have a high rate of cell division and high capacity for self-renewal. According to previous research, fetal HSCs and HPCs migrate to the liver where they mature before moving on to the fetal bone marrow [47]. As the embryo and fetus develop, hematopoiesis begins in several waves throughout the body, including the extraembryonic yolk sac, para-aortic area of the embryo, fetal liver and placenta, before finally homing to the bone marrow in the last weeks before delivery [48].

On embryonic day (E) 7, the yolk sac experiences primitive hematopoiesis, which is the first wave of hematopoiesis [49] as depicted in Figure 1. The yolk sac is an extra-embryonic membranous structure that is made up of two layers: an extra-embryonic mesoderm layer produced from epiblasts and a visceral endoderm layer from hypoblasts [50]. Most animals, including humans, do not have a yolk sac that surrounds the embryo. Instead, the embryo is linked to this membrane tissue [50]. This primitive hematopoiesis generates temporary hematopoietic cells with the ability to meet the requirements of the developing embryo, such as the primitive progenitor generation for erythroid (supplies the oxygen), macrophages (provide self-defense) and megakaryocytes (maintain the vascular system) [49,51]. However, this initial wave does not produce lymphoid cells. Large and nucleated primitive fetal erythrocytes are released into circulation, which differ from circulated adult erythrocytes in that the former is noticeably smaller and enucleated [52,53].

Intriguingly, diploid cells create proplatelets, which are lengthy extensions of cytoplasm that release platelets, in the E10.5 yolk sac [54]. The appearance of these cells is to release platelets into the circulating blood at about E11 and in fact, yolk sac-derived platelets are larger than adult bone marrow-derived platelets in size [55].

Moreover, at E8.5, the yolk sac marks the start of the second wave of hematopoiesis, which is characterized by the generation of lymphoid and erythromyeloid progenitors [56,57]. This wave is characterized as definitive hematopoiesis due to the differentiation of erythromyeloids into mature blood cells [49]. The third wave then starts around E10.5 and is identified by the production of HSPCs in the aorta-gonad-mesonephros (AGM) of embryos [58]. AGM is an embryonic tissue that develops from the mesoderm’s germ layer, containing the dorsal aorta, and is where the genitalia (which develops into the gonads) and mesonephros (which develops into the kidneys), are formed [59]. The AGM is a region of the body where HSCs are generated via the endothelial-to-hematopoietic transition from the hemogenic endothelium [48,60]. Nearly simultaneous to their emergence in the AGM is the appearance of definitive HSPCs in the placenta. Surprisingly, the identification of the HSPC niche within the placenta in both mice and humans has only been done within the last decade. The hematopoietic niche of the placenta within the labyrinth consists of not only the endothelial, perivascular and mesenchymal cells, but also placenta-specific syncytiotrophoblast cells [61]. In both species, the placenta provides a microenvironment that promotes the proliferation of HSPCs while preventing multilineage differentiation.

The placenta is a complex and diverse organ that is a high-yield source of HSPCs [62] and thus, can be a target for benzene toxicity and a representation of benzene’s effects in utero. Hematopoietic activity in the placenta occurs between E10.5–12.5 in which it plays a significant role in mouse and human blood cell development [63]. In vivo studies discovered that the placenta is capable of long-term HSPC repopulation [64,65]. Flow cytometry indicates that all HSPC cells at gestation day (GD) 12.5 exhibit phenotypes and a population comparable to the HSPCs in the fetal liver [64] which is a fetal tissue that has been shown to be a target of benzene toxicity [66,67]. Additionally, the placenta can produce an autonomous generation of HSPCs in isolation from the chorion [68]. These data indicate that the placenta has a role in HSPC generation between the dorsal aorta and the fetal liver, thus making it a target for benzene toxicity that could induce the first step of leukemogenesis. 

Last but not least, HSPCs will migrate via the blood circulation to the fetal liver at E12.5 where they will differentiate into different cell types in accordance with the hematopoiesis hierarchy [69] before colonizing in the bone marrow beginning at E.16.5. It is crucial to seed HSPCs in the bone marrow because erythropoiesis occurs primarily in the bone marrow during later stages of pregnancy [70]. This particular bone marrow will grow into the crucial hematopoietic organ during the last trimester of development as skeletal components start to ossify and bone marrow is formed inside bony cavities. The liver and spleen stop producing erythrocytes as the bone marrow now produces the majority of hematopoietic cells. Hence, it is noted that the developing fetal HSPCs need to reside in specific environments to acquire their properties; otherwise, the HSPC’s fate will be interrupted [71]. Analogous to the mouse, hematopoietic development in the human embryo occurs in a similar manner [72]. Therefore, exposure to various environmental carcinogenic agents is believed to interfere with the developmental programming of fetal HSPCs, leading to long-term hematopoiesis instability that promotes the development of hematological malignancies such as leukemia in the offspring.

## 4. Benzene Metabolism Linked to Fetal Toxicity via Maternal and Paternal Exposure

Current Occupational Safety and Health Administration (OSHA) standards in the United States sets the occupational exposure limit for benzene at 1 ppm (8 per 40 h workweek of total weight average allowable exposure limit, TWA) [73]. Meanwhile, the American Conference of Government Industrial Hygienists (ACGIH) sets the threshold limit value of benzene in the workplace at 0.5 ppm (1.6 mg/m^3^) [74]. The National Institute for Occupational Health and Safety (NIOSH) set the recommended exposure limit for 8 h of work at 0.1 ppm (0.32 mg/m^3^) [75]. Meanwhile, the Agency for Toxic Substances and Disease Registry (ATSDR) sets the maximum limit for benzene exposure at 0.009 ppm (0.028 mg/m^3^) per day which can have an acute effect and 0.003 ppm (0.009 mg/m^3^) per day which can have a chronic effect [76]. Short-term inhalation of benzene can cause headaches, dizziness, drowsiness, confusion and unconsciousness in humans. It is well known from numerous epidemiological studies conducted among workers that chronic exposure to benzene leads to adverse hematological effects. A cohort study including 32 people working in the shoe industry, who used benzene for between 4 months and 15 years and were exposed to concentrations of 15–30 ppm (49–98 mg/m^3^) outside working hours or 210–640 ppm (683–2080 mg/m^3^) during their work, found that the workers later developed pancytopenia with bone marrow changes [77]. The genotoxic capacities of benzene are due to its metabolites. Using the micronucleus (MN) assay, Pandey et al. [78] show that the metabolites of benzene, especially p-BQ, produce significant DNA damage. Animal studies also show that intermittent lifetime exposures to benzene at 980 mg/m^3^ are more tumorigenic than short-term high-level exposures at 3900 mg/m^3^ [79].

The co-existence of various pollutants in the environment and food chains is of considerably concern due to their synergistic impact on the environment and public health, especially if the joint toxicity of pollutants poses adverse health effects on humans [80]. In the context of benzene toxicity, reports concerning the synergetic effects of benzene with other carcinogens found in the soil or water are limited. For instance, formaldehyde and benzene are the two major indoor air pollutants due to their prevalence and toxicity. Formaldehyde has certain synergistic effects on benzene-induced cytotoxicity in peripheral blood due to ROS production and glutathione (GSH) depletion in spleen cells [81]. Female workers in the jewel-processing industry are more prone to DNA damage in the peripheral blood cells due to simultaneously being exposed to xylene, benzene and toluene [82].

Previous epidemiological, in vivo and in vitro studies have showed that benzene toxicity involves a number of hematological abnormalities involving different types of hematopoietic cell lineages such as aplastic anemia, myelodysplasia syndrome and leukemia [2,83]. On top of these epidemiological studies, many researchers have also revealed a causal effect of benzene and its metabolites on developing leukemia using both in vivo and in vitro models [6]. While most studies on leukemia have focused on adults, some epidemiologists have expanded their focus to examine the potential dangers of benzene exposure during pregnancy. The maternal and paternal routes are believed to be via the direct exposure to harmful agents during pregnancy or indirect exposure through the accumulation of toxins in the mother and subsequent transfer to the fetus through the placenta [84,85]. Table 3 shows an example framework for the biological mechanisms of childhood cancer in relation to exposure time windows via maternal and paternal routes.

Benzene’s closed structure of an aromatic ring makes it an unreactive chemical, hence its metabolism plays a crucial role in benzene toxicity [6]. In order to produce carcinogenic effects to the hematopoietic system, benzene must first undergo metabolism and subsequently produce more reactive and toxic metabolites, with some of the metabolites showing toxic effects on fetuses [7,87]. As shown in Figure 2, benzene may act via multiple modes of action targeting the developing fetal HSPCs niche, which include covalent binding, oxidative damage, errors in DNA repair pathways, chromosomal aberration and genetic damage, epigenetic modifications (DNA methylation as well as histone modification and chromatin remodeling) and placenta-mediated toxicity, all of which will be discussed in detail in Subtopic 5. With regard to benzene metabolism, initially, benzene undergoes the epoxidation process by the liver enzyme known as cytochrome (CYP) 2E1, resulting in benzene oxide. The majority of benzene oxide will be hydroxylated to create phenol, which will then be hydrogenated to create the reactive metabolite, HQ, by using P450 2E1 [88]. CYP 2E1 activity forms a mixture of metabolites from benzene, that diminishes the glutathione reserves, promotes oxidative stress, produces semiquinone free radicals and DNA lesions, triggers gene expression alterations as well as induces chromosomal aberrations related to leukemia [6,89]. The toxicity effects of benzene exposure on various experimental models are summarized in Table 4.

These metabolites will have varying degrees of ability to damage cells; for instance, HQ and catechol have considerably higher DNA-damaging abilities than t,t-muconic acid [105,106,107]. A previous study has shown that in C57Bl/6N mice, transplacental exposure to 200 mg/kg of benzene increases the level of ROS in the fetal liver and affects the number of hematopoietic colonies [67].

Furthermore, mouse bone marrow exhibits increased levels of nitrotyrosine products after exposure to benzene at a dose of 50 to 200 mg/kg [90]. Increasing levels of benzene in the fetus may trigger the production of detoxifying enzymes that would divert the metabolism of benzene from toxin-producing pathways that result in HQ and catechol. According to Ejiri et al. [108], fetal liver tissue has been found to upregulate several detoxification enzymes, including uridine 5-diphospho (UDP)-glucuronosyltransferase (UGTs) and glutathione S-transferase (GSTs). Benzene exposure at high levels, on the other hand, may inhibit the fetal body’s metabolizing enzymes through substrate inhibition [109,110].

An enzyme known as MPO can be found in abundance in bone marrow, as stated by [111]. This enzyme causes HQ to be activated, resulting in the production of 1,4-semiquinone, an intermediate metabolite that can later be converted into the more toxic 1,4-BQ. In contrast, an enzyme called NAD(P)H:quinone oxidoreductase 1 (NQO1) that can convert 1,4-BQ to HQ and subsequently detoxify 1,4-BQ, is also expressed by the bone marrow stroma cells, thus protecting the cells from BQ-induced oxidative stress [112]. People with NQO1 deficiencies are more severely affected by benzene toxicity than the general population. These findings suggest that further investigation on self-renewal and differentiation properties of HSCs and hematopoietic progenitors specifically in the fetus might progress our understandings of the benzene-induced in utero hematotoxicity and leukemogenicity mechanisms targeting the HSPCs niche.

## 5. Mechanisms of Benzene-Induced in Utero Carcinogenicity Involving Hematopoietic Stem Cells and Multilineage Progenitors

To date, there are a number of reports that have addressed the mechanism of in utero carcinogenicity as mediated following benzene exposure [66,67,113]. Nevertheless, the involvement of HSCs and multilineage HPCs in these mechanistic events remains elusive. Thus, this subtopic will highlight the mechanisms related to benzene-induced hematotoxicity and leukemogenicity with greater attention focusing on the in utero carcinogenicity and fetal HSPC niche. Studies related to this area of interest have been undertaken since the 1970s and still continue to be investigated in current scientific exploration. In the 1990s, studies conducted by several research groups found that embryos exposed to benzene during gestation show alterations in cell numbers for progenitor and hematopoietic precursors as well as increased granulopoiesis [91,92]. Radioactivity was also detected in the embryonic hematopoietic tissue of pregnant females who were exposed to radio-labelled benzene [93]. Meanwhile, in utero benzene exposure to pregnant mice during GD6 to 15 causes alterations in fetal tissue hematopoiesis that continue to persist for up to 6 weeks after birth as indicated by the affected myeloid and erythroid progenitor cells’ development [114]. Moreover, in another study, benzene was able to be detected in the fetal umbilical cord blood at the same or a greater level than in the mother’s blood, indicating the capability of benzene to cross the placenta and accumulate in the fetal-placenta unit [94]. However, despite these findings, there is a lack of studies that investigate in utero exposure to benzene involving HSCs and multilineage HPCs that are comprised of myeloid, erythroid and lymphoid progenitors. Previously, an experimental study had shown that yolk sac-derived HSCs are more sensitive to the cytotoxicity of HQ than the adult bone marrow-derived HSCs [104]. Thus, it is appropriate to assume that childhood leukemia could have developed during the fetal stage. According to an epidemiological study, maternal exposure to benzene during gestation can cause harmful effects on children such as the development of hematological malignancies that are commonly associated with childhood leukemia [115]. Figure 3 shows a summary of the studies addressing the incidences of childhood leukemia in relation to maternal exposure to benzene during pregnancy.

Based on the studies presented in Figure 3, the overall data suggest that acute lymphocytic leukemia (ALL) and AML are significantly associated with benzene exposure that occurs through maternal occupational exposures, usage of household products containing benzene and/or its metabolite and smoking during pregnancy. Moreover, for in utero exposure during gestation in mice, the same conclusion associated with increasing risk in developing leukemia has been reported. The risk of developing leukemia in childhood has been associated with exposure to various environmental variables, including solvents and paints [116]. According to another study, exposure to solvents at home may also be linked to childhood ALL [18]. Mothers that were exposed to paints were always linked to their children getting leukemia [117]. Hence, it would be wise to refrain from using paint in the home when pregnant or when the child is young [116]. Additionally, Swaen and Slangen [118] found a correlation between leukemia mortality and gasoline use, number of household cars, gas station and distance to the road being an assessment of petroleum indices. Thus, these results show that exposure to benzene has distinct toxicity effects depending on the different types of hematopoietic lineages. The mechanisms of the hematotoxic and leukemogenic effects of benzene and its metabolites are further discussed in the following subtopic.

### 5.1. Covalent Binding

The formation of reactive metabolites in the liver leads to covalent binding between macromolecules before they are transported via blood circulation to the organs [2]. For example, an adduct, known as S-phenylcysteine, is produced from the reactive metabolites of benzene, such as benzene oxide that was released from the liver into the blood in circulating hemoglobin and albumin [119]. Covalent binding of toxicants to cellular proteins can inhibit vital enzyme reactions, alter protein function, inhibit or stimulate membrane receptors as well as damage membrane proteins. It also can inhibit or alter protein synthesis or induce DNA mutations for nucleic acids such as ribonucleic acid (RNA) and DNA, leading to cellular dysfunction or carcinogenesis. Furthermore, covalent binding of toxicants to phospholipids can result in direct damage to cellular or organelle membranes and can trigger lipid peroxidation. In fact, for covalent binding, p-BQ and benzene oxide are significantly benzene-involved metabolites. A previous study reported that adult rats that were orally administered benzene showed lower levels of benzene oxide compared to BQ adducts in the bone marrow [95]. Elevated levels of benzene oxide and HQ adducts of hemoglobin and albumin have also been observed in workers subjected to benzene exposure [96].

Incubation of benzene metabolites with MPO enzymes extracted from the bone marrow or incubating benzene itself with microsomal proteins also causes the formation of adducts. Protein adduct’s formation will subsequently inhibit enzymes such as mtDNA polymerase, microsomal enzymes and topoisomerase II (Topo II) by inhibiting tubulin function leading to mitosis blockage. Moreover, previous in vivo studies reported the binding of benzene metabolites to DNA in the liver and bone marrow [120]. Meanwhile, previous in vitro studies demonstrated the binding of benzene to DNA and numerous feasible structures of benzene-induced DNA adducts involving 3′--hydroxy-1,N2-benzetheno-2′-deoxyguanosine and N7-phenylguanine [121]. However, most in vivo studies show that benzene metabolites binding to DNA occurred at extremely lower levels, suggesting that covalent binding itself does not completely explain benzene hematotoxicity and carcinogenesis; thus, other mechanisms should also be studied [122].

### 5.2. Oxidative Stress

Numerous benzene reactive metabolites have a high-level redox cycling potential, which typically results in the depletion of reducing equivalents, such as the GSH antioxidant and the generation of ROS, which leads to oxidative stress in cells [87]. As previously mentioned, MPO rapidly converts HQ to 1,4-BQ in the bone marrow. Sequential one-electron reductions of p- and oBQs can produce semiquinone (SQ•−), a free radical and then HQ. The unstable semiquinone causes it to cycle back to the other reactive metabolite, BQ and later forming the superoxide anion (O_2_•−) by passing an electron to oxygen in the bone marrow. In addition, by using superoxide dismutase, O_2_•− is dismutated to H_2_O_2._ Next, in the presence of iron (Fe^2+^), H_2_O_2_ generates •OH through the reaction known as the Fenton reaction. Nevertheless, the reduction of two-electron from BQ to HQ by NQO1 bypasses the formation of SQ•− and O_2_•− in the stroma cells. The deficiency of NQO1 increases the benzene toxicity and proves the significance of the balance in the BQ-HQ redox conversions following benzene toxicity. If p-BQ 2,3-oxide reacts with GSH, it will lead to the oxidation of p-BQ to 2,3-oxide and 1,2,4-trihydroxybenzene or to glutathionyl 1,2,4-trihydroxybenzene. Glutathionyl 1,2,4-trihydroxybenzene is extremely prone to autoxidation and redox cycling, which leads to the spontaneous production of O_2_•− and other radicals, thus playing a significant role in benzene-induced oxidative stress.

The muconaldehyde metabolite with an open ring structure may also raise oxidative stress due to its reactivity with the cysteine thiols of the redox pools in cells. Oxidative stress caused by benzene may impair macromolecules through several mechanisms such as the formation of 8-hydroxy-2′-deoxyguanosine (8-OHdG) from DNA base oxidation induced by BQ, HQ and 1,2,4-trihydroxybenzene, strand breaks of DNA and mutations, induction of homologous recombination and damage to the mitochondria [123,124,125]. For example, it was noted that with 1,4-BQ treatment in adult mouse bone marrow cells, malondialdehyde (MDA) levels increased through lipid peroxidation, demonstrating the ability of 1,4-BQ to cause oxidative damage and to induce DNA damage in HSPCs via lineage-dependent mechanisms [4]. It is well-known that exposure to benzene metabolites in murine fetal livers causes an increase in ROS as indicated by an increase in the fluorescent probe dichlorodihydrofluorescein diacetate (DCFDA) [113].

Previous studies have shown a significant increase in ROS production and significantly altered erythroid and myeloid colony numbers in fetal liver following in utero exposure to benzene in pregnant mice [97]. Moreover, the same exposure also causes an induction in embryonic expression of c-Myb as well as increased levels of total and phosphorylated Pim-1 [97]. Numerous cancers, including myelogenous leukemia, have an overexpressed signaling pathway including the c-Myb oncoprotein [126]. This transcription factor has a crucial role in numerous biological processes, including the control of cell proliferation, differentiation and apoptosis [126]. It is also crucial for hematopoiesis [127]. C-Myb is abundantly expressed in all proliferating immature cells and its transcriptional activity is subject to strict negative control [126]. The activity of c-Myb can be increased by Pim-1, a small (33 kDa) serine/threonine kinase, the altered activity of which has also been associated with leukemia [128].

### 5.3. Error in DNA Repair Pathways

In addition to oxidative stress, this review also looks into errors in DNA repair pathways as one of the benzene-induced pathways of in utero carcinogenicity. Transplacental exposure to environmental carcinogens in humans can cause DNA damage and induce somatic mutations in unborn children [129]. The development of DNA adducts, which can range in size from tiny and non-bulky lesions (such as N7-methylguanine) to massive and helix-distorting adducts (such as N7-aflatoxin-guanine adducts), was explored by [130] in 1999. The genotoxicity and hematotoxicity caused by benzene appear to be mostly dependent on DNA double-strand break (DSBs) repair [131]. DSBs are severe DNA lesions as a result of both damaged DNA strands that can cause cell death or a wide-rangeα of genetic changes such as large or small-scale deletions, loss of heterozygosity and chromosome, as well as translocations. This increases genome instability, which is a hallmark of cancer cells. Excision repair, such as base excision repair (BER) and nucleotide excision repair (NER), direct reversal (DR), mismatch repair (MMR) and DSBs repair pathways, such as homologous recombination (HR) and non-homologous end-joining (NHEJ), are examples of DNA repair pathways [132,133,134,135,136]. However, the focus of this review is on DSBs that are repaired by either NHEJ or HR [137]. In contrast to NHEJ, which is the dominant DSBR pathway during the rest of the cell cycle, HR primarily functions during the S and G2 phases of the cell cycle [136].

Chromosomal alterations, including inversions and deletions as well as enhanced somatic intrachromosomal recombination, have been correlated to increased NHEJ activity, which is a rapid but error-prone process [138]. DNA DSBs activate DNA-dependent protein kinase (DNA-PK), which later triggers NHEJ for the DSB repair mechanism [139]. NHEJ increases chromosome mutations and is error-prone, which contributes to cancer and genomic instability. According to Dewi et al. [13], exposure to 1,4-BQ mostly triggers DNA repair pathways through NHEJ, which has a notable impact on myeloid progenitor cells, indicating that the similar effect might be seen if the exposure was given to fetuses. The favored mechanism of DNA repair shifts with time in the developing fetus. However, in later stages of fetal development, the fetus prefers NHEJ to repair DNA damage [140]. As seen in Figure 4, there are a variety of ways in which fetal cells can react to DNA damage, any one of which could influence later health outcomes. During DNA replication, the presence of genetic lesions might result in an alteration to the DNA sequence, which may cause a mutation that can trigger cancer. Alternately, DNA damage may induce apoptosis in cells, which can lower the cell numbers in the developing conceptus [141].

Additionally, in vitro studies show that the benzene metabolites such as HQ, BQ and phenol dose-dependently induce γ-H2AX foci and DNA-PKcs, a hallmark of DNA DSBs [142,143,144]. As a result, NHEJ is promoted by benzene through increasing DSB and DNA-PKcs, which raises genomic instability. In the presence of DNA damage, dormant human HSCs favorably experience NHEJ for DNA repair rather than triggering apoptosis, accounting in part for the vulnerability of HSCs to benzene-induced bone marrow toxicity and leukemogenicity [145]. Moreover, the development of brain tumors and childhood leukemia have been linked to the apparent suppression of DNA repair proteins such as Topo II [146]. Fetal development depends on DNA replication with minimal errors and involves rapidly dividing cells. Disturbed Topo II resulting in increased dsDNA breaks leads to a higher frequency of NHEJ events and increases the fetus susceptibility to DNA misrepair and mutations [66]. A few studies have also addressed the problem of DNA repair during fetal development and environmental exposure to chemicals. For instance, fetal hematopoietic tissue obtained at GD16 was more sensitive to the formation of micronuclei (MN) than maternal hematopoietic tissue harvested at the same time when evaluating in utero genotoxicity of benzene [147]. This finding shows that in fetal development, benzene induces DNA damage repair, specifically DSB repair via the HR or NHEJ pathways that are maybe relatively insufficient during in utero development [147].

### 5.4. Chromosomal Aberration and Genetic Damage

Reactive oxygen and nitrogen species (ROS/RNS) produced during the progression of carcinogenesis lead to oxidative stress, which in turn causes DNA damage and other genotoxic consequences such chromosome aberrations and the formation of MN [148]. Exogenous chemical-induced genetic damage serves as both a crucial mechanism underlying the impacts of carcinogenic chemicals as well as a sensitive indicator of early health impairment. Consequently, genetic damage can be employed as a preliminary indicator for the adverse effects of carcinogens on health [149]. DNA cross-linking, DNA adduct formation, impaired DNA repair processes and sister chromatid exchange (SCE) are among the genetic harms allegedly driven on by benzene [150]. High levels of chromosomal alterations are linked to chronic exposure to benzene metabolites, as seen in cell cultures of human, including CD34^+^ progenitors [98]. Additionally, it is known that the benzene metabolites, HQ and 1,2,4-trihydroxybenzene cause chromosome breakage by inhibiting microtubule assembly, which leads to the formation of MN [151]. As depicted in Figure 5, MN are tiny extranuclear bodies that develop from acentric chromosomal fragments or complete chromosomes that lag behind at the anaphase of dividing cells and are not contained in the main nucleus during telophase. Instead, they are wrapped by the nuclear membrane and have a structure similar to that of the daughter nucleus, while being much smaller in size [152,153]. In MN, chromatids or chromosomes are formed as a result of chromosome segregation defects during anaphase, most commonly due to failure in the mitotic spindle, damage in the kinetochore, DSBs or hypomethylation of centromeric DNA [154].

The consequences of benzene exposure in adults were established in prior studies and they suggest a strong likelihood that the same impact would manifest in in the fetus. Transplacental cytogenetic effects of benzene in mice have been demonstrated in previous research. On day 14 of gestation, intraperitoneal injections of 0–874 mg/kg of benzene cause an increase in the frequency of MN in polychromatic erythrocytes isolated from the fetal liver and peripheral blood [99]. Additionally, Ciranni et al. [155] examined the transplacental effects of variety of metabolites in which only HQ results in the production of MN in mice fetal cells. Overall, exposure to carcinogenic agents during pregnancy is of particular concern for both the mother and child because it occurs during the period of their higher vulnerability to toxicants. Rapid organogenesis and growth during fetal development make this period more vulnerable to the harmful effects of toxic exposures [156].

### 5.5. Epigenetic Modification

Mutations have been established as the basis for the etiology of malignancies. They are of constitutional origin when inherited, or more commonly, of an acquired type when the origin is a genetic abnormality, occurring as a random event within the body or as a result of external mutagenic impact on the DNA. There are also non-mutagenic effects regulating the gene expressions called epigenetic effects. Gene expression regulation, proliferation and differentiation of stem cells and tumorigenesis have all been associated with epigenetics [157,158]. Furthermore, the epigenetic factors play a vital role in producing offspring. For the past 60 years, human genetic research has centered on the DNA as the heritable molecule that transfers the phenotype from the parents to the offspring [159]. Epigenetics is described as heritable changes in gene expression or cellular phenotype, which take place with no changes to the underlying DNA sequence mitotically and meiotically [160]. According to Lestari and Rizki [161], DNA methylation, histone modifications and chromatin remodeling are the primary epigenetic mechanisms, which also include gene expression regulation. DNA hypomethylation correlates with gene expression, while hypermethylation is correlated with gene silencing. A crucial process during early embryogenesis, genome reprogramming, is affected by DNA methylation and histone modifications [162].

#### 5.5.1. DNA Methylation

The most extensively studied epigenetic marker is DNA methylation, where the appropriate methylation of DNA is crucial for the development of the embryo. Gene expression and regulation, transcriptional silencing, genomic imprinting and X-chromosome inactivation are all related to DNA methylation [163,164]. Methylation occurs particularly through the action of the DNA methyltransferases (DNMT) family of proteins at the cytosine-phosphate-guanine dinucleotides (CpGs), which contain the 5-carbon position of cytosine. DNMTs are in charge for both the initiation of methylation and following methylation marks maintenance [165]. CpG dinucleotides appear in concentrations known as CpG-islands and despite their rareness, their presence in the genome can be discovered in the promoter region of nearly 50% of all human genes [166]. The promoter CpG-island regions are characteristically non-methylated in normal cells [166,167]. At the promoter regions, CpG hypermethylation triggers function loss, whereas global hypomethylation initiates genomic instability at repetitive sequences [167]. Disease phenotype is contributed to by these two types of alterations.

Changes that are often expressed in AML and other cancer tissues are due to aberrant DNA methylation models such as global hypomethylation, hypomethylation and gene-specific hypermethylation [168]. A preliminary study of workers exposed to benzene and control individuals was done by using DNA extracted from the peripheral blood mononuclear cells, in which the results present gender-specific methylation patterns for various genes and also changes in methylation at several CpG sites [169]. The induction of DNMT activity and in vitro-caused alterations were examined in an HaeIII DNMT-mediated methylation assay. The findings show that HQ and 1,4-BQ generate a statistically significant difference in global DNA hypomethylation compared with the control. These data indicate that HQ and 1,4-BQ can disrupt global DNA methylation and that 1,4-BQ-induced global DNA hypomethylation may have an impact via the DNMT activity inhibiting effects [100].

In addition, benzene exposure in in vitro studies has been reported to decrease global DNA methylation [170] along with a decrease in individual’s lymphocytes exposed to benzene at low dosages [168,171]. According to Philbrook and Winn [113], a significant decrease in global DNA methylation was observed in benzene-induced maternal bone marrow; however, no effect on global DNA methylation was noticed in fetal livers. Thus, this at least emphasizes the need for further research relating to epigenetic effects following in utero benzene exposure.

#### 5.5.2. Histone Modification and Chromatin Remodeling

Interestingly, normal and malignant hematopoiesis are known to rely on epigenetic processes such as histone modifications [172]. As such, epigenetic enzymes, notably histone methyltransferases, are recurrently mutated and dysregulated in hematologic malignancies [173]. Condensed heterochromatic or open euchromatic formation is established with the help of this modification. Histones are made up of amino-terminal extensions, a globular domain and the histone tail. Their DNA packaging-inducing components are abundant in the nucleus of eukaryotic cells and are high in the basic proteins lysine and arginine. Histone tail residues can be post-translationally modified (PTMs) in a variety of ways, including by acetylation, methylation, phosphorylation, sumoylation and ubiquitination. These PTMs alter the chromatin structure, allowing for the activation or inhibition of the underlying gene. Recent studies have demonstrated that histone core alterations also affect DNA replication, stemness and cell state modifications. Additionally, transcription is directly regulated by histone changes [174]. Acetylation defines the chromatin loosening and promotes replication and transcription, whereas methylation tightens DNA and prevents access to a variety of enzymes. These modifications depend on the exact covalently bound group and the specific amino acid involved [101].

Modified proteins have been found in the bone marrow and liver of mice following administration of 155 and 800 µg/kg benzene [101]. The results prove the extensive binding of benzene metabolites to proteins and their high reactivity. Comprehensive studies of the histones have shown that they are broadly attacked by reactive benzene species, subsequently causing the occurrence of multiple modified sites within a single histone, in which the degree of carbon-14 of benzene incorporation occurs independently of the specific histone [101]. Moreover, an enhanced global histone H3 trimethylated at lysine 4 (H3K4me3) modification was observed in benzene-exposed workers and is positively associated with the benzene exposure level and hematotoxicity [102]. In vitro studies revealed that the H3K4me3 modification is in complex with specific damage response and repair (DDR) genes and regulates gene expression in response to HQ treatment [102]. H3K4me3 integrates a variety of signaling pathways, including transcription initiation, elongation and RNA splicing [175]. Collectively, these observations provide new insight into the epigenetic mechanism, particularly the involvement of specific histone modifications with the regulation of cellular responses that might occur in the developing fetus.

### 5.6. Placenta-Mediated Toxicity

Benzene exposure can influence the proliferation of both B and T cells, lessen the host endurance to infection and generate chromosomal aberrations [176]. These detrimental health effects due to benzene exposure have been well documented in the adult stage. However, there is a scarcity of studies evaluating the clinical findings and undesirable health effects of benzene exposure in the fetus and children. Although the literature on the health outcomes of benzene in children is scant, emerging studies show that benzene exposure through placental barrier can cause deleterious health effects in both fetus and children. Limiting the fetal exposure to harmful chemicals found in the mother’s blood is one of the placenta’s roles. However, this defense is insufficient, allowing the benzene to pass through the placenta and into the vascular system of the developing fetus [177]. According to Prouillac and Lecoeur [178] as well as Burton and Fowden [179], the efficiency of placenta screening varies depending on physiological and chemical changes that occur in the placenta throughout pregnancy. For instance, changes in the expression of genes closely related to the transporting protein and the enzymes involved with xenobiotic metabolism, the thickness of the placenta, the electrical gradient between the maternal and fetal circulation that affects the conversion of the molecular charge will influence fetal development [178,179].

The fetus has the ability to metabolize carcinogenic chemicals [180]. The evidence shows that the metabolism appears to be activated in the placenta. Previous research on human placentas in vitro using precarcinogenic benzo[a]pyrene exposure perfusion demonstrated that the chemical not only can pass through the placenta tissue into the fetus but also causes alterations in the DNA [181]. In addition, carcinogen exposure during pregnancy damages fetal genetics and elevates the chances of getting cancer [182].

Maternal exposure to leukemogenic factors in early pregnancy may increase the DNA instability of the hematopoietic system, genetic susceptibility to cancer and oncogenic lesions during the fetal stage in the hematopoietic system, which can result in the development of childhood leukemia in their offspring [20]. The placenta, which serves as the fetus and mother’s interface throughout pregnancy, is crucial to the programming of the fetus [183,184]. Specifically, trophoblast cells in the placenta create the partition between maternal and fetal blood. The chorionic villi of the placenta are lined with a bilayer of trophoblast during the first trimester and they are submerged in maternal blood pool [185]. Transport between the mother and the fetus occurs through this trophoblast barrier.

Trophoblast barriers that have been exposed to oxidative stress-causing chemicals can release factors that harm DNA in human embryonic stem cells [186,187] and human fibroblasts [188,189]. DNA damage signals and tumor necrosis factor alpha (TNF-α) are at least two of the signals released by the trophoblast bi-layer. For cell death to occur, both of these signals must be present; neither one by itself is adequate. A cytokine with pleiotropic effects, TNF-α plays a crucial regulatory role in controlling cell growth, differentiation and programmed cell death. Studies have shown that in the placenta, uterus and embryos of both humans and animals, TNF-α and its receptor are expressed [184,186]. Miscarriage is correlated with high TNF-α levels [188,189,190].

## 6. The Origin of Hematological Diseases from In-Utero Benzene Exposure

Although benzene is widely used in the industrial setting, it has toxic effects on the haematology, reproduction and neurology systems as a cancer-causing carcinogen. Many workers are exposed to high levels of benzene in the workplace air. It is stated that hundreds of leukemia cases caused by benzene exposure have been reported from some industries, such as the charcoal, shoes and paint industry [191]. In occupationally exposed adults, benzene causes AML and possibly other blood cancers including ALL, chronic lymphoid leukemia (CLL), multiple myeloma and non-Hodgkin lymphoma [192,193]. Fetuses and children might also be exposed to benzene either through air pollution, sources near residences (such as car repair facilities or gas stations), maternal workplace exposure during pregnancy or home use of products containing benzene [194]. Despite the adequate data linking benzene to leukemia in adults, it is less certain whether there is a correlation in exposed children suggesting the need for further investigation focusing on maternal benzene exposure.

Previous studies of neonatal dried blood spots, found that ALL is likely initiated in utero, where identified ALL-related leukocyte chromosomal translocations were identified in children who later developed leukemia [195]. Benzene metabolites induce instability in genomic and chromosomal aberrations [51], thus it is likely that they could cause the characteristic translocations seen in ALL. Moreover, [196] reported that muconic acid, a degradation product of benzene, can be found in 29 pregnant women living close to natural-gas hydraulic fracturing sites in which the muconic acid in their urine is 3.5 times greater than that of the general population. A blood cancer such as leukemia occurs due to uncontrolled cell division of myeloid and lymphoid progenitor cells and subsequently unable to proceed the differentiation process to produce mature and functional blood cells [103]. To date, LSC is identified to be formed from HSCs and/or HPCs that have undergone mutations in which they have maintained or reacquired the capacity for indefinite and uncontrolled proliferation via accumulated mutations [197,198]. In addition, exposure of mouse embryonic yolk sac HSCs and adult bone marrow HSCs to HQ were reported to promote apoptosis, inhibited cell proliferation, differentiation and the clonogenic potential of both HSCs [104].

Childhood leukemia has distinctive features compared with leukemia in adults. In white populations aged 0 to 14 years old, the main subtypes are ALL (80%) and AML (15%) cases, respectively [199]. Both subtypes are believed to progress via a first initiating event in utero (for example the TEL-AML1 gene fusion in newborns has a prevalence of about 1% while it is found in 25% of ALL cases) followed by further postnatal genetic changes [199].

## 7. Conclusions and Future Remarks

In conclusion, accumulating evidence suggests that benzene and its reactive metabolites are risk factors for in utero carcinogenicity as previously demonstrated through epidemiological and experimental studies. Previous studies have also shown that benzene and/or its metabolites may act as hematotoxic and leukemogenic agents via multiple modes of action targeting the adult HSPCs niche, which include covalent binding, oxidative damage, error in DNA repair pathways, chromosomal aberrations and genetic damage, epigenetic modifications and placenta-mediated toxicity leading to modifications of the HSPCs and probable carcinogenesis particularly leukemogenesis. However, there is a lack of evidence linking the mechanism of benzene toxicity with in utero carcinogenicity targeting the HSPC niche. Thus, extensive research is needed to bridge the gap between clinical observations and epidemiological studies in the context of in utero carcinogenesis induced by benzene exposure. This is a fundamental need as the developing fetus is highly susceptible to chemical exposure that could occur through occupational and environmental-linked maternal exposure as well as through lifestyle practices by the mother. Therefore, the research outcomes focusing on in utero carcinogenicity by benzene exposure could provide a better understanding of the potential molecular signature of cancer stem cells that could be useful for targeted cancer treatment, particularly for childhood cancer, as well as for risk assessment and policy decisions concerning chemical and carcinogen exposure affecting pregnant women.

## Figures and Tables

**Figure 1 ijms-24-06335-f001:**
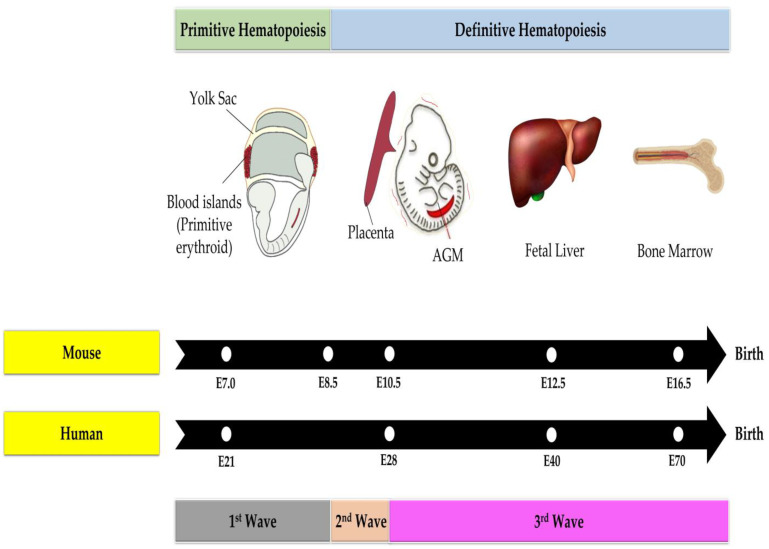
Hematopoiesis in fetal phase for mouse and human. Abbreviations: AGM: aorta-gonad-mesonephros; E: embryonic day.

**Figure 2 ijms-24-06335-f002:**
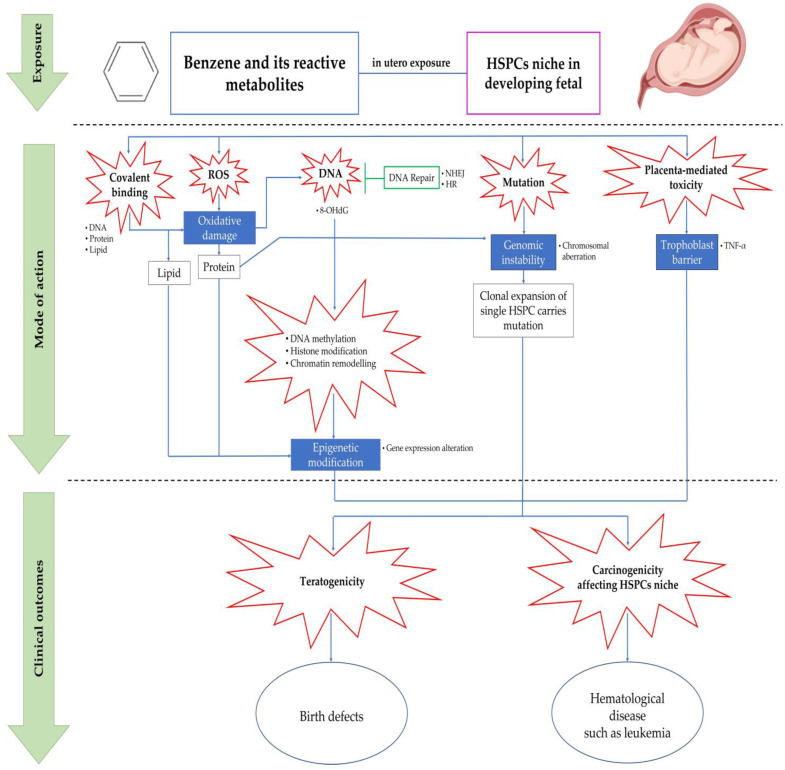
In utero exposure, mode of action and clinical outcomes of benzene and its metabolites on HSPCs niche in developing fetal. Abbreviations: DNA: deoxyribonucleic acid; HR: homologous recombination; 8-OHdG: 8-hydroxy-2′-deoxyguanosine; mEH: microsomal epoxide hydrolase; NHEJ: non-homologous end-joining; ROS: reactive oxygen species; TNF-α: Tumor Necrosis Factor alpha.

**Figure 3 ijms-24-06335-f003:**
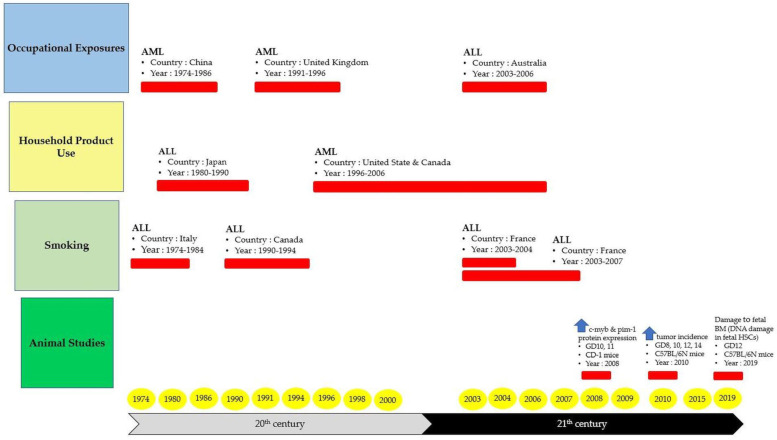
Summary of the studies addressing the incidences of childhood leukemia in relation to maternal exposure to benzene during pregnancy. Abbreviations: ALL: acute lymphocytic leukemia; AML: acute myeloid leukemia; DNA: deoxyribonucleic acid; GD: gestation day; HSCs: hematopoietic stem cell.

**Figure 4 ijms-24-06335-f004:**
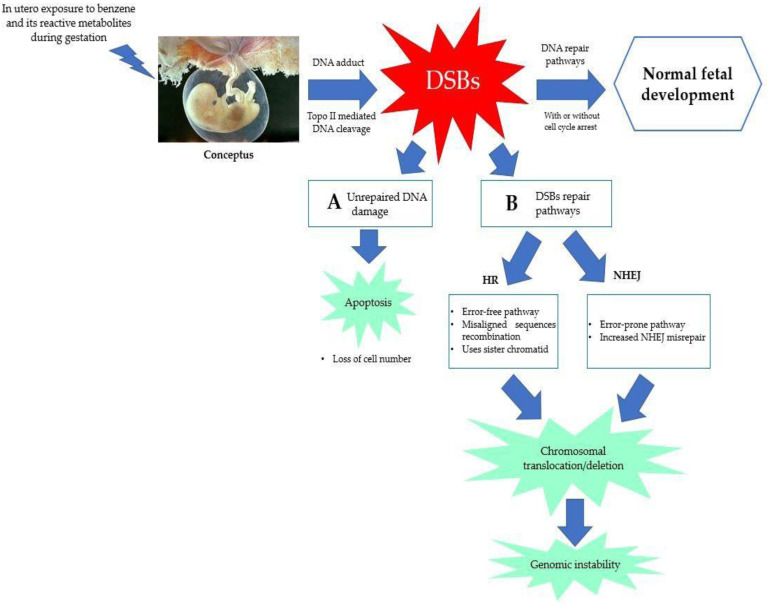
Potential cellular events and health outcomes on fetus from DNA damage during in utero development. Benzene metabolites effectively inhibit Topo II causing Topo II-mediated DNA cleavage and produce DNA adducts, leading to the formation of DNA strands with DNA adduct generating DSBs. (**A**) A programmed cell death (apoptosis) will occur as a result from enormous and unrepaired DNA damage. (**B**) The DSBs repairs pathways occur through HR and NHEJ. HR-mediated DNA repair utilizes the sister chromatid exchange pathway as template for accurate repair, producing error-prone DNA pathway. Meanwhile, NHEJ-mediated DNA pathway is an error-prone repair that frequently leads to misrepaired DSBs. Abbreviations: DSBs: double-strand breaks; HR: homologous recombination; NHEJ: non-homologous end-joining; Topo II: topoisomerase II.

**Figure 5 ijms-24-06335-f005:**
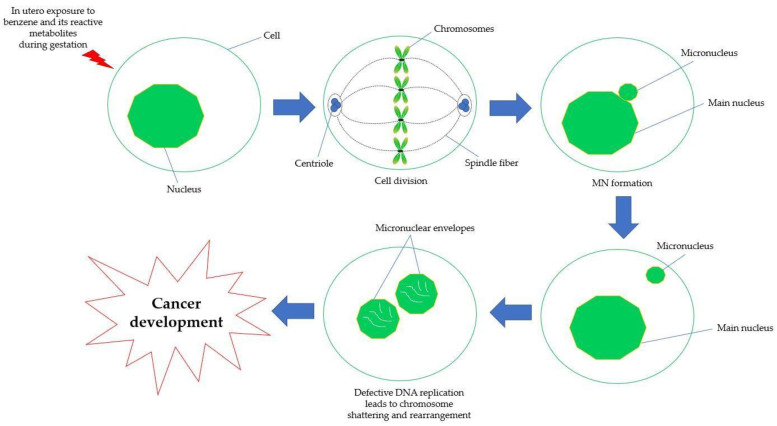
Micronucleus formation in the cell of in utero exposure to benzene and its metabolite. Abbreviations: DNA: deoxyribonucleic acid; MN: micronuclei.

**Table 1 ijms-24-06335-t001:** List of the key characteristics of carcinogens.

1. Is electrophile or can be metabolically activated to electrophiles. 2. Is genotoxic. 3. Alters DNA repair and causes genomic instability. 4. Induces epigenetic alterations. 5. Induces oxidative stress. 6. Induces chronic inflammation. 7. Is immunosuppressive. 8. Modulates receptor-mediated effects. 9. Causes immortalization. 10. Alters cell proliferation, cell death or nutrient supply.

Adapted from [16]. Abbreviations: DNA: deoxyribonucleic acid.

**Table 2 ijms-24-06335-t002:** Summary of the article selection criteria used to construct this review.

No	Category	Description
1	Journal Databases	Google Scholar PubMed
2	Inclusion Criteria	Articles contain the keywords “benzene”, “fetal”, “in utero”, “hematopoietic stem cell and progenitors”, “carcinogenicity”, “oxidative stress”, “epigenetic”, “chromosome aberration”, “hematological disorders”. Articles that are linked directly to in utero exposure of benzene and its mechanism of toxicity, hematopoietic stem cells and progenitors as well as development of hematological disease. The searches for articles were not refined by publishing date, authors, author affiliations, journals or the impact factors of the journals. Quantitative articles that provided measurable data that uncovered trends and patterns in in childhood cancer statistic due to benzene exposure Qualitative articles that provided insights into the problems and ideas or hypotheses underlying the toxicity in fetal following maternal exposure to benzene.
3	Exclusion Criteria	None
4	Types of articles	87 original research articles 103 review articles 1 conference paper 1 commentary paper 6 webpages 1 thesis

**Table 3 ijms-24-06335-t003:** Framework for biological mechanisms of childhood cancer in relation to exposure time windows via maternal and paternal routes.

**Exposure Time Windows**	**Periconception** **(Pre- and Around Conception)**	**Intrauterine (Gestational)**	**Postnatal**
**Maternal**	Germ cell mutation or DNA damage within ovaries	Transplacental: (1) Maternal exposure during pregnancy to carcinogens either directly or indirectly (2) Release of toxins in the mother and subsequent transfer to the developing fetus	Breast feeding: (1) Release of toxins into breast milk through direct or indirect maternal exposure to toxicants (2) Release of already accumulated toxins in maternal tissues to breast milk Direct: (1) Exposure of the child to carcinogenic agents brought by the mother
**Paternal**	Germ cell mutation or DNA damage within spermatozoa Binding of accumulated toxins in seminal fluid to spermatozoa before/during fertilization	Transplacental (1) Release of toxins from paternal seminal fluid during intercourse to the developing fetus	Direct: (1) Exposure of the child to carcinogenic agents brought by the father

Adapted from [86].

**Table 4 ijms-24-06335-t004:** Toxicity effects of benzene exposure on various experimental models.

Benzene or Metabolites	Study Design	Dose or Concentration	Experimental Model	Toxicity Effects	References
1,4-BQ	In vitro	0, 1.25, 2.5, 5 μM	ICR mice	Induce concentration-dependent cytotoxicity & apoptosis in BM cells, ↓total counts of Sca-1^+^, CD11b^+^, Gr-1^+^ & CD45^+^ cells, ↓clonogenicity in 1,4-BQ-treated cells	[12]
1,4-BQ	In vitro	0, 5, 7, 12 μM	ICR mice	↓colony-forming capacity of the myeloid progenitor at 1.25 & 2.5 μM, ↑expression HoxB4 level at all concentrations & ↑Bmi-1 expression level at 5 μM, ↑GATA3 expression level at 2.5 µM	[14]
Benzene	Epidemiology	–	Pregnant women	↑risk of childhood ALL	[18]
BQ	In vitro	6.25, 9.375 12.5, 15.625 μM	Murine CD-1	↓c-kit^+^Lin-Sca-1-Il7rα-cell population in BQ-treated, ↓Topo IIα activity in concentration-dependent, ↑γH2AX levels at 12.5 µM BQ exposure	[66]
Benzene	In vivo (i.p.)	200, 400 mg/kg (GD8, 10, 12, 14, 16)	Pregnant C57Bl/6N mice	↑numbers of CFU-E, BFU-E, CFU-GM & CFU-G colonies at 200 mg/kg, ↓numbers of CFU-M colonies in hematopoietic tissue of GD16 C57Bl/6N fetuses at 200 mg/kg, ↑numbers of CFU-E & CFU-G colonies at 400 mg/kg benzene on GD8, 10, 12, 14, ↑ROS production in fetal liver	[67]
Benzene	In vivo (i.p.)	50, 100, 200, 400 mg/kg	B6C3F1 mice	↑nitration of tyrosine residues in bone marrow proteins from 50 to 200 mg/kg	[90]
Benzene	In vivo (inhalation)	10 ppm (GD6–15)	Pregnant Swiss Webster mice	↑alteration in cell numbers for progenitor & hematopoietic precursor	[91]
Benzene	In vivo (inhalation)	5, 10, 20 ppm (GD6–15)	Pregnant Swiss Webster mice	↓numbers of circulating erythroid precursor cells, ↑numbers of hepatic hematopoietic blast cells & granulopoietic precursor cells, ↑granulopoiesis.	[92]
Radio- labelled benzene	In vivo (inhalation)	–	Pregnant C57BL mice	Radioactivity detected in the embryonic hematopoietic tissue	[93]
Benzene	Epidemiology	–	Pregnant women	Benzene detected in fetal umbilical cord blood at the same or greater level than in the mother’s blood, ↑benzene accumulation in the fetal-placenta unit	[94]
Benzene	In vivo (oral)	50, 100, 200, 400 mg/kg	Fischer (F344) rats	↑BQ adducts of Hb and bone-marrow proteins with dose-dependent manner	[95]
Benzene	Epidemiology	≤31, >31 ppm	Workers	↑levels of BO & HQ adducts of Hb and albumin	[96]
1,4-BQ	In vitro	0, 1.25, 2.5, 5, 7, 12 μM	ICR mice	↓GSH level, ↓SOD activity, ↑MDA level, ↑PC level, ↑DNA damage in BM cells (↑DNA in tail % at 7 & 12 μM as well as ↑tail moment at 12 μM), ↑DNA damage in myeloid & pre-B lymphoid progenitors at 2.5 µM, ↑DNA damage in the erythroid progenitor at 5 µM 1,4-BQ, ↑in tail moment at 7 µM & 12 µM 1,4-BQ exposure for all progenitors	[4]
Benzene	In vivo (i.p.)	200 mg/kg (GD8, 10, 12, 14)	Pregnant CD mice	↑oxidative stress in fetal tissue from embryos, ↑ROS sensitive fluorescent probe DCFDA, ↑expression of fetal Pim-1, ↑Pim-1 phosphorylation, ↑c-Myb, ↑phosphorylated p38-MAPK, ↓protein levels of Iҡßα	[97]
HQ	In vitro	0, 5, 10, 15, 20 μM	Human	↑frequency of the specific chromosomal aberrations in CD34^+^ bone marrow cells	[98]
Benzene	In vivo (i.p.)	0, 109, 219, 437, 874 mg/kg (GD14)	Pregnant Swiss Webster mice	↑frequency of MNPCE in fetal liver & fetal peripheral blood cells at 219 to 874 mg/kg, ↑frequency of MNPCE in maternal bone marrow cells at 437 & 874 mg/kg	[99]
HQ & 1,4-BQ	In vitro	5, 10, 25, 50 μM	L02 cell line	↑global DNA methylation, ↓DNMT activity	[100]
Benzene	In vivo (i.p.)	155, 800 µg/kg	B6C3F1 mice	↑protein adducts in bone marrow & liver	[101]
Hydroquinone	In vitro	0, 0.1, 1.0, 10.0 µM	Worker’s blood	↑global H3K4me3 modification & hematotoxicity, ↓white blood cells count, ↓neutrophils count, ↓lymphocytes count, ↓monocytes count	[102]
Muconic acid	Epidemiology	–	Pregnant women	↑muconic acid in urine of pregnant women living close to natural-gas hydraulic fracturing sites	[103]
Hydroquinone	In vitro	0, 1.25, 2.5, 5.0 µM	SPF Kunming mice	↓proliferation & differentiation as well as ↓colony formation of both YS-HSCs & BM-HSCs, ↑apoptosis of both YS-HSCs & BM-HSCs	[104]

Abbreviations: ALL: acute lymphocytic leukemia; BFU-E: burst-forming unit erythroid; BM: bone marrow; BM-HSCs: bone-marrow hematopoietic stem cells; BQ: benzoquinone; BO: benzene oxide; CFU-E: colony-forming unit erythroid; CFU-GM: colony-forming unit granulocyte-macrophage; CFU-G: colony-forming unit granulocyte; DCFDA: dichlorofluoroscein diacetate; DNMT: DNA methyltransferases; DNA: deoxyribonucleic acid; GD: gestation day; GSH: glutathione; Hb: hemoglobin; HQ: hydroquinone; ICR: imprinting control region; i.p: intraperitoneal; MAPKs: mitogen-activated protein kinases; MNPCE: micronucleated polychromatic erythrocytes; PC: protein carbonyl; ppm: part per million; ROS: reactive oxygen species; SOD: superoxide dismutase; SPF: special pathogen free; Topo IIα: topoisomerase IIα; YS-HSCs: yolk sac hematopoietic stem cells; ↑ increase; ↓ decrease.

## Data Availability

Not applicable.

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
