# Peer review of "Linking Benzene, in Utero Carcinogenicity and Fetal Hematopoietic Stem Cell Niches: A Mechanistic Review"

_ijms, 2023, doi:10.3390/ijms24076335_

Round 1

Reviewer 1 Report

This review relates to the mechanisms of in utero carcinogenicity by benzene, with HSPCs as targets. This is an important point to ask.

It is an extensive review, which covers the hematopoietic system (adult and fetal), benzene metabolism linked to fetal toxicity, the different mechanisms of benzene-induced in utero carcinogenicity involving HSPCS, and finally the origin of hematological diseases from in utero benzene exposure.

I have some major concerns about this review:

1.     The quality of English, which often makes the paper difficult to read. The manuscript needs to be reviewed and edited by a native English.

2.     As it stands this review lacks a common thread, and the reader who is interested on links between pediatric or adult cancers and exposure to carcinogens during fetal life has hard time. For instance, it would benefit from more focused reflection on the fact that benzene can have a deeper effect on fetal HSPCs: what are the specificities of fetal HSPCs that make them a preferential target of benzene? What about microenvironment during fetal life (fetal HSPCs make an initiatory journey through different niches before getting to bone marrow)

3.     The paragraph 3 “Hematopoietic system in the bone marrow niche” is incomplete, whether at the level of adult bone marrow niche or fetal hematopoiesis. One cannot reduce the bone marrow niche to vascular network: I agree that this network is important, but what about mesenchymal cells? All over the world many teams work on the dialog between HSPCs and MSCs during normal and pathological situations.

Regarding fetal hematopoiesis: in the mouse embryo, after their emergence in the floor of aorta in the AGM region, HSPCs not only migrate to fetal liver but also to placenta (quoted by the authors), where they get amplified. This is of importance, especially in light of placenta as a barrier. In that respect, cells of placenta microenvironment (vascular, MSCs) can also be affected by benzene, and the dialog between HSPCs and these later distorted, and could induce first steps of leukemogenesis? Also, it is of importance to describe human fetal hematopoiesis.

One main characteristic of fetal hematopoiesis that certainly makes fetal HSPCs more sensitive to the effects of benzene or other carcinogens than adult counterpart is the fact that HSPCs are in active division all along fetal life, while they are resting in the adult bone marrow niche (from 3 to 4 weeks after birth for mice).

4.     I may have missed the point, but I did not see any comment or discussion about the dose-effect of benzene: it seems clear that if one is subjected to high concentrations of benzene during adult or fetal life, the risk of carcinogenesis is very high. What are these concentrations, and what happens when people are subjected to low but permanent doses of benzene? What are the synergetic effects of benzene with other carcinogens found in the soil or water (cadmium, arsenic…)? Are there reports on that specific point?

Minor concerns: 

-       Survey methodology: is this necessary?

Author Response

Hi, good evening

Please see the attachment below. Thank you.

Reviewer 2 Report

The author reviewed and proposed the links  between benzene exposure, in uterus carcinogenesis with hematologic malignancies. 

1. It is better for authors to have a table with the toxicity effects of benzene exposure from references for readers' convenience.

Author Response

(The authors gave the same response as above.)
